# Multiplex Recombinase Polymerase Amplification Assay for Simultaneous Detection of *Treponema pallidum* and *Haemophilus ducreyi* in Yaws-Like Lesions

**DOI:** 10.3390/tropicalmed5040157

**Published:** 2020-10-06

**Authors:** Michael Frimpong, Shirley Victoria Simpson, Hubert Senanu Ahor, Abigail Agbanyo, Solomon Gyabaah, Bernadette Agbavor, Ivy Brago Amanor, Kennedy Kwasi Addo, Susanne Böhlken-Fascher, Jonas Kissenkötter, Ahmed Abd El Wahed, Richard Odame Phillips

**Affiliations:** 1Department of Molecular Medicine, College of Health Sciences, Kwame Nkrumah University of Science and Technology, Kumasi AK-448, Ghana; ahor@kccr.de; 2Kumasi Centre for Collaborative Research in Tropical Medicine, Kwame Nkrumah University of Science and Technology, Kumasi AK-312, Ghana; a.agbanyo@yahoo.com (A.A.); solomongyabaah53@gmail.com (S.G.); agbavor@kccr.de (B.A.); phillips@kccr.de (R.O.P.); 3Bacteriology Department, Noguchi Memorial Institute of Medical Research, University of Ghana, Accra GA-337, Ghana; ssimpson@noguchi.ug.edu.gh (S.V.S.); iamanor@noguchi.ug.edu.gh (I.B.A.); kaddo@noguchi.ug.edu.gh (K.K.A.); 4Division of Microbiology and Animal Hygiene, Georg-August University, D-37077 Goettingen, Germany; susanne.boehlken-fascher@agr.uni-goettingen.de (S.B.-F.); jonas.kissenkoetter@uni-goettingen.de (J.K.); ahmed.abd_el_wahed@uni-leipzig.de (A.A.E.W.); 5Institute of Animal Hygiene and Veterinary Public Health, University of Leipzig, D-04103 Leipzig, Germany

**Keywords:** recombinase polymerase amplification, *Treponema pallidum*, *Haemophilus ducreyi*, molecular diagnostics, point-of-care test

## Abstract

Yaws is a skin debilitating disease caused by *Treponema pallidum* subspecies *pertenue* with most cases reported in children. World Health Organization (WHO) aims at total eradication of this disease through mass treatment of suspected cases followed by an intensive follow-up program. However, effective diagnosis is pivotal in the successful implementation of this control program. Recombinase polymerase amplification (RPA), an isothermal nucleic acid amplification technique offers a wider range of differentiation of pathogens including those isolated from chronic skin ulcers with similar characteristics such as *Haemophilus ducreyi* (*H. ducreyi*). We have developed a RPA assay for the simultaneous detection of *Treponema pallidum* (*T. pallidum*) and *H. ducreyi* (TPHD-RPA). The assay demonstrated no cross-reaction with other pathogens and enable detection of *T. pallidum* and *H. ducreyi* within 15 min at 42 °C. The RPA assay was validated with 49 clinical samples from individuals confirmed to have yaws by serological tests. Comparing the developed assay with commercial multiplex real-time PCR, the assay demonstrated 94% and 95% sensitivity for *T. pallidum* and *H. ducreyi*, respectively and 100% specificity. This simple novel TPHD-RPA assay enables the rapid detection of both *T. pallidum* and *H. ducreyi* in yaws-like lesions. This test could support the yaws eradication efforts by ensuring reliable diagnosis, to enable monitoring of program success and planning of follow-up interventions at the community level.

## 1. Introduction

Yaws, most common of the three endemic non-venereal treponemal infections is caused by *Treponema pallidum* sp. *pertenue* [1,2]. Yaws presents as lesions of the skin (papilloma), which ulcerates and progresses to cause destructive lesions of bone and cartilage leading to chronic disfigurement and disability when left untreated. The transmission is via direct (non-sexual) contact with fluid exudate from the lesion of an infected person [3]. The disease mostly affects children living in poor communities of West Africa, Latin America, Asia, and the Pacific where access to healthcare is limited [3]. Estimates in 2012 show that over 89 million people are affected with Yaws in 15 countries in Africa, Southeast Asia, and the Pacific region [4].

The WHO aimed to achieve total eradication of yaws by 2020 through a comprehensive large-scale treatment strategy, called the Morges strategy [5]. This strategy consists of an initial mass drug therapy with oral intake of azithromycin in endemic communities followed by an active surveillance every 6 months to actively detect and treat remaining cases and their contacts. Pilot implementation of this strategy in endemic countries such as Ghana, Congo, Papua New Guinea, Vanuatu, and the Solomon Islands have shown promising results, with the need for scale-up [6]. However, the success of the Morges strategy is hinged on the selection of the most appropriate diagnostic assay at each stage of the eradication effort to ensure a complete halt in the transmission of yaws [7]. The occurrence of *Haemophilus ducreyi*, as a common cause of chronic skin ulcers similar to yaws in endemic areas also compounds the challenges of diagnosing yaws [8,9]. *H. ducreyi* is a common cause of genital ulcers called chancroid, and it is now increasingly recognized as a cause of non-genital skin ulcers in children in developing countries [8]. In patients with yaws-like lesions, differential diagnosis other than those cause by *T. pallidum pertenue* is important for accurate reporting and management of cases.

Traditional serological tests such as Treponema Pallidum Particle Agglutination assay (TPPA) and Rapid Plasma Reagin test (RPR) have remained the common diagnostic tool for yaws, over the years [1]. These tests are encumbered with issues of low sensitivity and specificity hence are not appropriate for use during the follow-up stage of the eradication strategy [10]. The Dual Path Platform (DPP) Syphilis Screen-and-Confirm assay (Chembio Diagnostic Systems, Inc., Medford, NY, USA), which enables the simultaneous detection of antibodies to treponemal and non-treponemal antigens, is accurate for the confirmation of clinically suspected cases. This point-of-care (POC) test has increased sensitivity in high seropositive individuals and low sensitivity in individuals with low antibody titre [10]. In addition, issues of co-infections cannot be overlooked as reactive syphilis serology caused by latent yaws has been observed in ulcers with *H. ducreyi* [8]. The WHO recommends the use of molecular diagnostic tests as part of the case definition in yaws eradication effort [7]. This highlights the need for a sensitive and specific diagnostic test in confirming yaws suspected cases even after mass drug administration (MDA). 

Nucleic acid amplification techniques, especially polymerase chain reaction (PCR), can address the issues of diagnosis of yaws, due to their high clinical sensitivity and specificity to *T. pallidum* and *H. ducreyi*. This DNA-based diagnostic tests could play a central role in the yaws eradication strategy by ensuring effective diagnosis and surveillance after MDA [7,8,11]. However, PCR techniques can only be applied in a reference laboratory and not at the point of care (POC) in endemic communities. As such, the need for diagnostic techniques that do not involve costly equipment and can be used in remote or resource limited areas have been indicated [12]. The use of isothermal amplification techniques such as Loop-mediated amplification (LAMP), for on-site screening has been developed as a possible POC test for chronic skin ulcers caused by *T. pallidum* and *H. ducreyi* [13,14,15]. These LAMP assays for *T. pallidum* and *H. ducreyi* demonstrated high sensitivity (85−100%) and specificity (85−100%). Nevertheless, the use of unextracted DNA reduced the sensitivity of *T. pallidum* [15]. LAMP assays are generally performed at 60−65 °C and results are obtained after 30−60 min., LAMP uses four to six primers which are not easy to design especially for highly variable pathogens under field conditions. In contrast, recombinase polymerase amplification (RPA), has emerged as a better field-deployable molecular diagnostic tool, since it yields readily readable results within a shorter turnaround time (less than 15 min) and it is less complicated to run at a lower temperature (37–42 °C) compared to techniques such as LAMP [16,17]. RPA have been performed at low resource settings in a mobile suitcase laboratory coupled with field-friendly DNA extraction techniques such as SpeedXtract (Qiagen, Hilden, Germany) for effective diagnosis of infectious diseases [18,19,20]. The technique opens the door to extending the application of DNA amplification in the field or point-of-need to differentiate between pathogens commonly isolated from chronic skin ulcers [21]. This isothermal nucleic acid amplification technique can be used at all stages of yaws eradication efforts, especially during the follow-up phase, monitoring of azithromycin resistance as well as identifying individuals with *Treponema pallidum* and *H. ducreyi* co-infection in low resource settings. In this study, we have developed a RPA assay for the simultaneous detection of *T. pallidum* and *H. ducreyi* (TPHD-RPA) in DPP confirmed yaws patients.

## 2. Materials and Methods 

### 2.1. Clinical Samples

Samples were obtained from an ongoing skin neglected tropical diseases (NTDs) project (Yaws, Leprosy and Buruli ulcer) community outreach program being conducted in Wassa Amenfi East, Upper Denkyira East, Upper Denkyira West, Upper West Akyem, Akwapim North and Sekyere afram plains districts of Ghana. In brief, clinically suspected yaws patients were first screened with immunochromatographic assay (SD Bioline Syphilis 3.0 RDT kit, Standard Diagnostics Inc., Suwon, South Korea) for the qualitative detection of antibodies of all isotypes (IgG, IgM, IgA) against *Treponema pallidum* (TP). Positive samples by this test were then tested with the DPP Syphilis Screen-and-Confirm assay (Chembio Diagnostic Systems, Inc., Medford, NY, USA), according to manufacturer instructions for the detection of antibodies to treponemal and non-treponemal antigens. Swab samples were taken for both ulcers and wet papilloma, while scab was removed from dry or closed papilloma from patients who were DPP positive. Samples were placed in a sterile patient ID pre-labeled cryotubes containing 700 μL cell lysis solution (CLS, Qiagen, Hilden, Germany) and transported to Kumasi Center for Collaborative Research in Tropical Medicine (Kumasi, Ghana) and Noguchi Memorial Institute for Medical Research (Accra, Ghana) depending on the proximity of the district to the reference laboratory for further molecular analysis. 

This study used archived samples with consent from study participants for future use of samples. Approval was also sort from the Committee for Human Research Ethics and Publication (Ref: CHRPE/AP/122/17), School of Medical Sciences, Kwame Nkrumah University of Science and Technology (Kumasi, Ghana). 

### 2.2. DNA Extraction

DNA extraction was performed using the Gentra Puregene Tissue Kit (Qiagen GmbH, Hilden, Germany) according to manufacturer’s instructions. Briefly, samples stored in lysis buffer were vortexed for 2 min, swab sticks removed from the solution and then centrifuged for 1 min at 13,000 rpm to pellet the cells. Supernatant was carefully discarded. A total of 300 µL of CLS was added to the pellet after which 15 µL of lysozyme was added and mixed by inverting 25 times. It was then incubated at 37 °C for 30 min and at 80 °C for 5 min to lyse the cells. Proteins were precipitated by adding 100 μL of protein precipitation solution (Qiagen, Hilden, Germany) to the sample, vortexed 20 s and centrifuge at 13,000 rpm to pellet. Supernatant was transferred into a pre-labelled 1.5 mL tube (Eppendorf AG, Hamburg, Germany), containing 300 μL isopropanol and 2 μL glycogen. This mixture was mixed by inverting gently 50 times and then centrifuged for 1 min at 13,000 rpm to pellet DNA. The supernatant was carefully discarded and 300 µL of 70% ethanol added to wash the pelleted DNA and then centrifuged for 1 min. Alcohol supernatant was carefully discarded and DNA air dried for 5−30 min. The extracted DNA was resuspended in 100 µL DNA Hydration Solution (Qiagen, Hilden, Germany), vortex for 5 s and incubated at 65 °C for 1 hr. Extracted DNA was stored at –20 °C till it was used for *T. pallidum* and *H. ducreyi* RPA (TPHD-RPA) assay and the multiplex real-time PCR (qPCR).

### 2.3. T. pallidum and H. ducreyi Multiplex qPCR Assay

The multiplex qPCR assay was performed with RealCycler^®^ universal kit (Progenie molecular, Valencia, Spain) for the detection of the *PolA* and *HhdA* specific genes for *T. pallidum* and *H. ducreyi,* respectively, on a Bio-Rad CFX96 Real-time PCR detection system (Bio-Rad Laboratories, Paris, France) according to manufacturer instructions. Briefly, 14 µL of RealCycler^®^ Universal AmpliMix (qPCR mastermix reagent) was pipetted into reaction tubes and 6 µL of DNA template was used to yield a reaction volume of 20 µL. Positive and negative controls were also included in each run. Cycling condition of PCR are as follows one (1) cycle of initial denaturing at 95 °C for 15 min followed by 45 cycles of 95 °C (denaturing) for 5 s, 60 °C (annealing) for 30 s and 72 °C (extension) for 30 s. 

### 2.4. T. pallidum and H. ducreyi RPA (TPHD-RPA) Assay 

#### 2.4.1. Molecular Standard

Molecular DNA standard of 600 bp, made up of 300 bp *T. pallidum PolA* gene (GenBank: U57757.1, range 1836−2135) and 300 bp of *H. ducreyi hemolytic cytotoxin HhdA* gene (GenBank: U32175.1, range 5031−5330) was synthesized by GeneArt (Invitrogen, Darmstadt, Germany). The number of DNA molecules per microliter was determined using the equation described previously [22]. Serial dilutions between 10^6^ and 10^0^ of the molecular standard were prepared. 

#### 2.4.2. TPHD-RPA Primers and Probes

*PolA* and *HhdA* genes of *T. pallidum* and *H. ducreyi* respectively, were used as target sequences for RPA detection. RPA primers and probes were designed in accordance with the TwistDx manufacturer’s recommendations (TwistDx Limited, Maidenhead, United Kingdom). Three forward primers (FPs), three reverse primers (RPs) and one exo probe (P) each for *T. pallidum* and *H. ducreyi* (Appendix A) were designed by using the Molecular Evolution Genetics Analysis version 7 software (MEGA, State College, PA, USA). In silico BLAST analysis of these oligonucleotide demonstrated high specificity to the *T. pallidum* and *H. ducreyi*. All the primers were produced by Eurofins Genomics (Ebersberg, Germany) and the exo probe by TibMolBiol (Berlin, Germany). 

#### 2.4.3. TPHD-RPA Assay 

Preliminary screening of three forward primers, three reverse primers and one exo probe combinations were tested with 10^5^ molecules/5 µL of molecular standard in singleplex RPA assay for *T. pallidum* and *H. ducreyi* using the TwistAmp Exo kits according to the manufacturer’s instructions in a final reaction volume of 50 µL (TwistDX, Cambridge, UK). The reaction for each target contained 2.1 µL each of 10 µM forward and reverse primers, 0.8 µL each of 10 µM probe, 29.5 µL rehydration buffer, 8.0 µL DNase-free water, 2.5 µL of 280 mM Magnesium Acetate (MgOAc). Reaction tubes were incubated in an isothermal fluorometer (T8-ISO Axxin Pty Ltd., Victoria, Australia). Using the T8-ISO Desktop application, we programmed the fluorometer to detect the lowest dilutions that met criteria for distinguishing positive samples from negative controls as indicated previously [21]. All tests were run at 42 °C for 15 min with mixing after 4 min. To be considered positive, a sample required either a gradient or an amplitude (plateau florescence) of at least 900 mV over a 40 s sliding window during the amplification phase of the test (5−15 min). The best primer and probe combination for *T. pallidum* and *H. ducreyi* (Table 1) were later produced as mix for subsequent RPA assay using TwistAmp Exo kit, with minor modification of manufacturer’s instructions. The reaction volume was 50 µL and contained 2.5 µL each of TP and HD 10 µM primer/probe mix, 29.5 µL rehydration buffer, 8.0 µL DNase-free water, 2.5 µL of 280 mM Magnesium Acetate (MgOAc). Five microliters of DNA extract were used.

#### 2.4.4. Analytical Singleplex RPA Assay Sensitivity and Specificity 

The sensitivity of the singleplex RPA assay was evaluated using as serial dilations (10^6^ to 10^0^ copies/µL) of TP/HD DNA molecular standard in eight separate reaction in triplicate. The threshold time and copies of DNA molecules detected was plotted, and a semi logarithm (log) regression was calculated using PRISM 8.0 software (GraphPad Software, San Diego, CA, USA). The specificity of TPHD-RPA assay was evaluated using genomic DNA of other pathogens (Appendix A).

#### 2.4.5. Clinical Performance of TPHD-RPA Assay

The clinical performance of the TPHD-RPA assay was evaluated with forty-nine (49) DPP positive yaws suspected samples. The result of the TPHD-RPA assay was compared with the results *T. pallidum* and *H. ducreyi* multiplex qPCR assay. 

#### 2.4.6. Statistical Analysis 

All data were entered in Microsoft Excel 2016 while GraphPad Prism v.8 (GraphPad software, San Diego, CA, USA) was used for data analysis and drawing graphs. The limit of detection (LOC) of both qPCR and singleplex RPA assay was determined using probit regression analysis (GraphPad Prism v.8). Descriptive information of patients recruited such as percentages, frequency, median and interquartile ranges were obtained using descriptive statistics. Contingency tables were used to determine the sensitivity, specificity and the predictive values of the TPHD-RPA assay, using qPCR results as the gold standard.

## 3. Results

### 3.1. Analytical Sensitivity of Singleplex RPA Assay and Specificity of TPHD-RPA Assay

The nine oligonucleotide combinations for each assay were screened with the molecular DNA standard. Primers for TP (TP_RPA_FP3 and TP_RPA_RP2) and HD (Hd_RPA_FP3 and Hd_RPA_RP1) when combined TP_RPA_P1 and Hd_RPA_P1 respectively (Table 1) were able to amplify down to ten (10) of DNA molecules (Figure 1). 

The gradient or an amplitude florescence of at least 900 mV signal over a 40 s sliding window for both singleplex RPA assay for 10^6^ to 10^3^ copies of DNA molecular standard was at 5−6 min, 6−7 min for 10^2^ copies, and 8−9 min for 10 copies (Figure 1).

Using probit analysis, the limit of detection of the singleplex RPA assay at 95% probability was 11 and 6 copies of DNA molecular standard for *T. pallidum* and *H. ducreyi* respectively (Figure 2). The assay showed no cross-reactivity when tested with a panel of 29 genomic DNA of bacterial and parasite species (Appendix A).

### 3.2. Clinical Performance of TPHD-RPA Assay

Forty-nine sample from DPP positive participants were included for the evaluation of the TPHD-RPA assay for *T. pallidum* and *H. ducreyi* with majority being males (N = 31, 63%). The median age of participants was 10 years. They presented with ulcers (N = 35, 71%), papillomas (N = 12, 25%) and both (N = 2, 4%). Using qPCR, 10 lesions were identified as *T. pallidum* lesions, 14 as *H. ducreyi* lesions, 7 lesions coinfected with *T. pallidum* and *H. ducreyi,* and 18 (37%) negative for both pathogens.

The TPHD-RPA assay also correctly identified all samples containing only one pathogen and five samples (71%) being coinfected. Two (4%) samples positive for both pathogens by qPCR, were either positive for either *T. pallidum* or *H. ducreyi* and not both. Using qPCR as the gold standard test, the diagnostic sensitivity of the TPHD-RPA assay for *T. pallidum* and *H. ducreyi* was 94% and 95%, respectively (Table 2). Kappa coefficients (κ), ranging from 0.95 to 1 for the detection of *T. pallidum* and *H. ducreyi* demonstrate an excellent diagnostic agreement between qPCR and our developed TPHD-RPA. Additionally, excellent agreement between qPCR and TPHD-RPA (κ = 0.9) was demonstrated for lesions coinfected with both pathogens. All negative qPCR samples were negative by TPHD-RPA assay demonstrating 100% specificity (Table 2). 

## 4. Discussion

Eradication of yaws is a major priority by WHO with the detection of *H. ducreyi*, an essential component in the management of yaws patients. In yaws endemic communities, clinical diagnosis combined with serological testing is the sole means of diagnosing suspected yaws cases. Nucleic acid amplification techniques can help overcome the limitations of serological tests, by enabling the determination of whether yaws like lesions are caused by either *T. pallidum* or *H. ducreyi* or both. Hence these tests are needed in the WHO yaws eradication strategy [7,8,10,11].

In this study, we developed a real-time recombinase polymerase assay for the simultaneous detection of *T. pallidum* and *H. ducreyi* (TPHD-RPA) in yaws-like lesions. The developed TPHD-RPA assay was highly specific to the target organisms while the limit of detection of singleplex RPA assay was 11 and 6 copies of DNA for *T. pallidum* and *H. ducreyi* respectively. These analytical sensitivities are comparable to multiplex qPCR whose sensitivity is 100 and 1 copy for *T. pallidum* and *H. ducreyi* respectively (RealCycler^®^ universal TPHD-U TPHD-G protocol V.4). The turnaround time of 15 min for the developed TPHD-RPA assay, suggest that this assay could provide rapid molecular diagnosis of *T. pallidum* and *H. ducreyi* lesions as well as a distinction between these two pathogens in areas where co-infection has been reported to enhance case management [8,9].

Polymerase chain reaction provides high sensitivity and specificity for diagnosing yaws and *H. ducreyi* lesions [7,8,11]. Operating this technique at the point of care is very challenging. This is because of its requirement for sophisticated laboratories, equipped with expensive equipment such as a thermocycler [12], which can cost more than 15 times as much as isothermal fluorometers required for the performance RPA. RPA performance requires less costly equipment and can be performed in less resource settings in a mobile suitcase laboratory [18,20]. As such, TPHD-RPA could be an alternative molecular diagnostic tool which can be essential in the yaws eradication strategy. 

Recently, the use of Loop-mediated amplification (LAMP) for on-site screening has been proposed as a future point of care test for chronic skin ulcers caused by *T. pallidum* and *H. ducreyi* (TPHD-LAMP) [13,14]. These TPHD-LAMP assays had a limit of detection of 300−600 DNA copies with a diagnostic performance of 85−92% sensitivity and 85−96% specificity [14]. A *T. pallidum* LAMP assay has also been developed with 100% sensitivity and specificity reported [15]. The sensitivity of this singleplex assay was reduced (67%) when evaluated with unextracted DNA samples. In our study, the developed TPHD-RPA assay showed 86−100% sensitivity and 100% specificity. These diagnostic performances of both TPHD-RPA and TPHD-LAMP suggest the potential use of these amplification techniques as an alternative to PCR for confirming clinically suspected yaws cases. However, RPA has an added advantage of rapidly amplifying nucleic acid target at 39–42 °C within a short test time of 15 min compared to LAMP which requires a higher temperature of 60−65 °C with a turnaround time of 30 min to an hour [13,14,15]. All critical RPA reagents thus enzymes/proteins and deoxy-nucleoside triphosphates are lyophilized into a single pellet which ensures the stability of reagents at room temperature for at least six months, reduces contamination, facilitates easy transportation of reagents and simplifies RPA applicability in low resource settings [23,24]. In addition, RPA assays have been evaluated with field friendly extraction techniques such as simple boiling of samples [24,25,26], SpeedXtract (Qiagen, Germany) [18,27,28] and GenoLyse DNA extraction Kit (Hain Life science, Nehren, Germany) [21] with little or no effect on RPA diagnostic performance. These DNA extraction techniques are less labor intensive with only one/two manipulation or pipetting steps and do not require sophisticated laboratory settings and hence applicable in low resource settings or mobile suitcase laboratories [18,27,28]. This suggests that the developed TPHD-RPA can be used with such DNA extraction techniques in low resource settings. 

This study used DNA extracted with Gentra Puregene Tissue commercial Kit (Qiagen GmbH, Hilden, Germany) in a well-equipped laboratory. As such the novel TPHD assay should be evaluated with the above-named field friendly extraction procedures. This would help ascertain the full diagnostic performance of the developed assay. 

## 5. Conclusions

We have developed a simple novel TPHD-RPA assay which can enable rapid differential detection of *T. pallidum* and *H. ducreyi* in yaws-like lesions. The diagnostic performance of the TPHD-RPA suggests its potential to increase access to molecular diagnosis of yaws, especially in low resource yaws endemic communities. This test, when combine with an effective field friendly extraction technique could support all yaws eradication efforts by ensuring reliable diagnosis, enabling monitoring of eradication effort success and planning of follow-up interventions at the community level.

## Figures and Tables

**Figure 1 tropicalmed-05-00157-f001:**
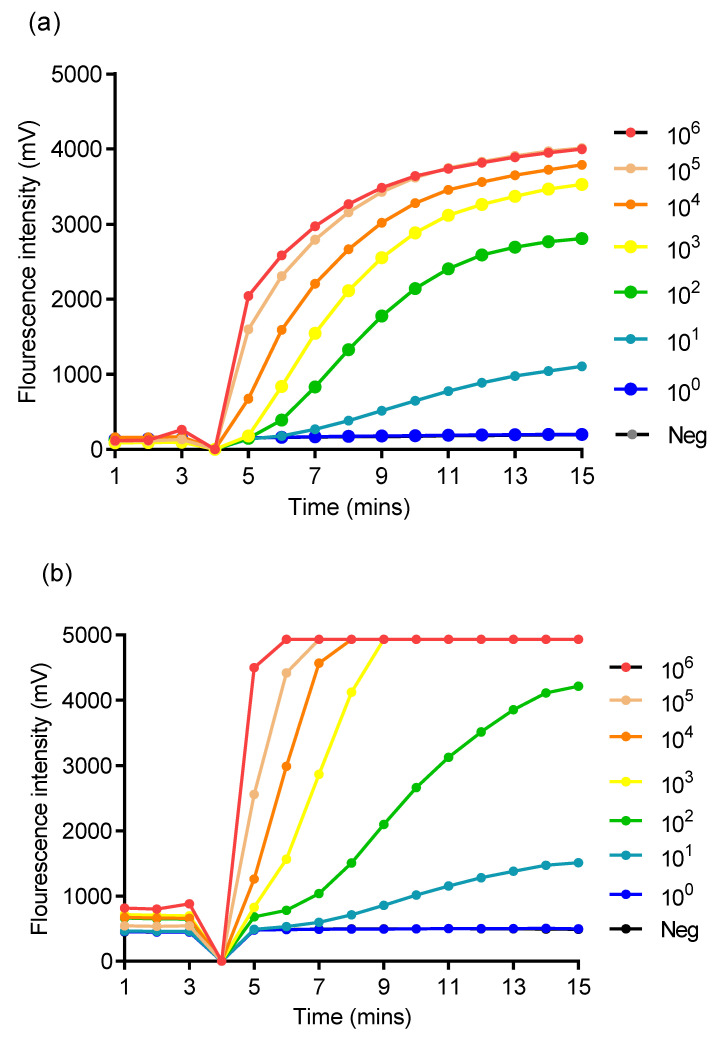
Sensitivity analysis of the singleplex RPA assay using synthetic molecular DNA standard. The assay showed a sensitivity of 10 copies/reaction in *T. pallidum* and *H. ducreyi.* RPA reaction tubes were removed and quick vortexed/mixed four minutes after state of amplification and hence no florescence recorded. Sensitivity test for *T. pallidum* (**a**) and *H. ducreyi* (**b**). “Neg” is negative control or no template control (NTC).

**Figure 2 tropicalmed-05-00157-f002:**
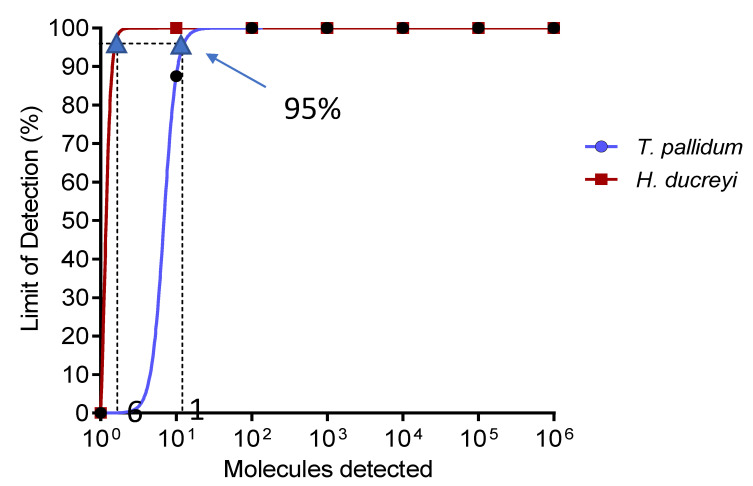
Probit regression analysis. The limit of detection at 95% probability is 11 and 6 DNA molecules for *T. pallidum* and *H. ducreyi*, respectively. CI: confidence interval.

**Table 1 tropicalmed-05-00157-t001:** RPA Primer and Probe for *T. pallidum* and *H. ducreyi* Detection.

Name	Sequence	Amplicon Size
TP_RPA_FP3	GTAACACTAATGTCGAGACTGAAAAGGAGTGC	144 bp
TP_RPA_RP2	GAATGATGAGACGCTCACACTTGTTATGC	
TP_RPA_P1	GTATCTGCATCTGCTGTGCAGGATCCGGCA (BHQ2-dT) (X-dSpacer) (ROX-dT)GTCCAAGCTGTCATG-PH	
Hd_RPA_FP3	TACTCAGGCAACGGATACCCAACATGCA	169 bp
Hd_RPA_RP1	GAGGTAAATCAGGCTGTTACAGGTCATTTA	
Hd_RPA_P1	TACGCCTAAATCGTTAACTGCGGGATTAGG (BHQ2-dT) (X-dSpacer) (FAM-dT)AGATGGCCATGGTAG-PH	

FP: Forward primer, RP: Reverse primer, P: probe, FAM-dT and ROX-dT: thymidine nucleotide carrying Fluorochrome; dSpacer: tetrahydrofuran basic site mimic, BHQ2-dT: thymidine nucleotide carrying Black Hole quencher, PH: 3′ phosphate group.

**Table 2 tropicalmed-05-00157-t002:** Diagnostic Performance of TPHD-RPA Assay as Compared to Multiplex qPCR for Each Pathogen.

Target Organism	Samples Category	Sensitivity %	Specificity %	PPV %	NPV %	N	TPHD-RPA	qPCR	
		(95% CI)	(95% CI)	(95% CI)	(95% CI)			pos	neg
***T. pallidum***	All samples	94	100	100	97	49	pos	16	0
		(73−100)	(89−100)	(81−100)	(85−100)		neg	1	32
	Samples containing a single organism	100	100	100	100	42	pos	10	0
		(72−100)	(89−100)	(72−100)	(89−100)		neg	0	32
	Samples containing both pathogens	86	100	100	97	39	pos	6	0
		(49−100)	(89−100)	(61−100)	(84−100)		neg	1	32
***H. ducreyi***	All samples	95	100	100	97	49	pos	20	0
		(77−100)	(88−100)	(84−100)	(83−100)		neg	1	28
	Samples containing a single organism	100	100	100	100	42	pos	14	0
		(78−100)	(88−100)	(78−100)	(88−100)		neg	0	28
	Samples containing both pathogens	86	100	100	97	39	pos	6	0
		(49−100)	(89−100)	(61−100)	(85−100)		neg	1	32

CI: confidence interval, PPV: Positive predictive value, NPV: Negative predictive value, n: Number of samples tested, pos: positive, neg: negative.

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
