# Peer review of "Multiplex Recombinase Polymerase Amplification Assay for Simultaneous Detection of Treponema pallidum and Haemophilus ducreyi in Yaws-Like Lesions"

_tropicalmed, 2020, doi:10.3390/tropicalmed5040157_

Round 1

Reviewer 1 Report

I believe the manuscript should be accepted as is

Author Response

Authors thank reviewer for this comments and support to this manuscript.

Reviewer 2 Report

Review Report

“Multiplex Recombinase Polymerase Amplification Assay for Simultaneous Detection of Treponema pallidum and Haemophilus ducreyi in yaws-like lesions” by Frimpong et al.

A brief summary

The authors presented a valuable study with potential to deal with an important public health issue and to aid the WHO-initiative in total eradication of jaws.  

General comments 

The study presents a development of a simple novel assay for simultaneous detection of two pathogens causing skin ulcers. It should be mentioned that H. ducreyi commonly causes genital ulcers (chancroid) and it is increasingly recognized as a cause of nongenital skin ulcers in children in certain developing countries.

The authors should offer a broader overview of the issue. Importantly, I find that the authors should give a rationale for the study/testing in the clinical setting of very limiting sources in the light of the fact that the first-line treatment of the ulcers caused by both pathogens is a single dose of azithromycin (e.g. for chancroid infection with H. ducreyi “should be started promptly, without waiting for test results.” MSD Manual).

Furthermore, the authors should offer a balanced view of the field (e.g. other diagnostic methods – LAMP, DPP) and refrain from self-evaluation of importance of their findings.

A great care should be taken when using the abbreviations in the manuscript. Please, be consistent when using the abbreviations and do not “over-use” the abbreviations, since the manuscript might be difficult to read (PLOS ONE - “Non-standard abbreviations should not be used unless they appear at least three times in the text.”) When the abbreviation is once explained, it is not necessary to explain it several time (e.g. TPDH RPA, DPP, LAMP, TP, HD).

The study presents a “simple novel TPHD-RPA assay enables the rapid detection of both T. pallidum and H. ducreyi in yaws-like lesions.” The “Materials and Methods” section should provide enough detail that a competent worker could repeat the experiment.

The authors could consider some other articles employing RPA, in order to present their study more clearly and thoroughly (e.g. Lei, R., Wang, X., Zhang, D. et al. Rapid isothermal duplex real-time recombinase polymerase amplification (RPA) assay for the diagnosis of equine piroplasmosis. Sci Rep 10, 4096 (2020). https://doi.org/10.1038/s41598-020-60997-1 or Cabada MM, Malaga JL, Castellanos-Gonzalez A, Bagwell KA, Naeger PA, Rogers HK, Maharsi S, Mbaka M, White AC Jr. Recombinase Polymerase Amplification Compared to Real-Time Polymerase Chain Reaction Test for the Detection of Fasciola hepatica in Human Stool. Am J Trop Med Hyg. 2017 Feb 8;96(2):341-346. doi: 10.4269/ajtmh.16-0601.)

Great care should be taken in the comma setting throughout the manuscript, since omission of the comma in not just few sentences in the text leads to misunderstandings.

Some concerns and suggestions regarding different segments of the manuscript are below.

Introduction

The introduction should better summarize relevant research to provide context in a way to explain the importance and a need for a novel assay.

Can be the purpose of the work stated in the form of the hypothesis or a question?

Line 66 - unclear: “low in…”

Line 85-89 – please rewrite “Studies…”

Line 89 – “This isothermal…” – repetitive

Materials and Methods

Please, include required information about the manufacturer of all used reagents and equipment (e.g. RDT, DPD PPS, lysis buffer vs. CLS), CLS, BioRad, Mega7).

Take care that the abbreviation for the units is in scientific writing always in the singular (e.g. not secs, mins).

Line 108 – NMIMR – Country?

Line 109 – “ongoing” – repetitive. “For use…” – not important for this study.

Line 121 – “centrifuge…” – rpm? “at high pellet” – unclear.

Line 122 – omitting “eppendorf"?

Line 124 – “high speed” – rpm?

Line 126 – “DNA [something missing]”

Line 132 – please, write the gene names correctly. hgbA vs. HhdA? Please, include the primer sequences. Reaction volume? Primer Mix volume? Amplification in triplicate?

Line 139 – was the molecular standard a mixture? Equals TP/HD DNA molecular Standard (line 168)? Relation to “DNA Standard” in line 187?

Line 150 – “Additional file: Table1” was not available for review purposes. Please include the sequence data for all primers.

Line 155 – please state clearly, that nine singleplex reaction were tested in the preliminary screening. Please include information on volumes of different reagents.

Line 160 – “A combined…” - please include a reference and/or explain the procedure.

Line 161 – “…as mix” – Please, include all data about the assay. That is important. Which “modification”?

Line 167, 171 – RPA assay or assays?

Line 177-181 – please rewrite.

I am not a statistician, so I cannot evaluate the appropriateness of the used statistical methods.

Results

The authors may consider presenting data on qPCR.

The authors should present the data on performance of the novel TPDH-RPA assay in graphical form (i.e. positive for TP, for HD and for both).

The authors may consider presenting the information from Table 1 in Materials and Methods section.

Table 1 is not self-understanding. Please explain the abbreviations. Were here primers FP3+RP2 and FPS+RP1 presented?

Line 191-195 – I guess that here is not the legend to the Figure 1.

Line 191 – “using…” Where is the data shown? Exponential amplification was not achieved at all concentrations.

Line 193 – “The start…” – could the threshold value be depicted on the Figure 1?

Line 194 – “one DNA…” – unclear. Relates to?

Figure 1 - Personally, I find the term “10^0” unusual. What is the difference between “10^0” and “Neg”? The data from “neg” are not visible. “10 copies/reaction” – unclear.

Line 122 – please omit “…while"

Line 203 – “Cross-reactivity” testing has to be mentioned in Materials and Methods section. Please include all necessary information about this testing in Materials and Methods section. I think, that the Table 2 should be a supplementary Table.

Figure 2 – please rescale the figure, so the most important segment of the X-axis is easily readable. From the Figure 2 it is not readable that “the limit of detection at 95% probability is 11 and 6 DNA molecules”.

Table 2 – it does not make sense to include the third column – the data is same for all tests. Some “sources” are missing.

Line 213 – please rewrite.

Line 214-216 – please, do not repeat the results in text and Table 3 (erroneously “Table 2”).

Table 3 – I would suggest to present the data just in text, since the table gives rather scarce information.

Line 219 – 229 and Table 4 – please, do not repeat the results in text and table. If the authors decide to present the data in the table, just few results should be also mentioned in the text. Please, explain kappa coefficient.

Line 222-223 – a sample cannot be infected.

Table 4 – what is the meaning of the “*”? Why is the data for positive samples positive for both pathogens differs (“7” in text and second column vs. “6” in 3rd and 4th column)?

Discussion

The Discussion should connect to the Introduction by addressing the problems. In the field, what is the rationale to use a singleplex or multiplex RPA test?

Please offer balanced discussion (e.g. Line 233 – “major priority”, Line 246 – “very essential”, Line 255 – “essential”)

Line 241 - “This analytical…” – belongs this segment to some other section?

Line 243 – “The diagnostic…” – From which data is this conclusion drawn?

Line 245 – What is “RT-RPHD RPA”? Please avoid abbreviation where possible.

Line 258 – “This TPHD-LAMP…” – I find that this information should be mentioned in earlier sections of the manuscript.

Line 259 – “in contrast…” - please rewrite.

Line 265 – “Also, all…” – which reagents? I find that this information should be mentioned in earlier sections of the manuscript.

Line 273 – “…GenoLyse…” – I find that this statement does not “belong” to the concluding remarks of the Discussion. The issue of the DNA extraction should be definitively elaborated earlier in the text.

Other sections

Line 300 – it should be written “assays”.

Line 371 – it should be written “Böhlken-Fascher”.

Reviewer 3 Report

General

The manuscript describes the use of a real-time recombinase polymerase assay for the detection of T. pallidum and H. ducreyi, and is well written except for a few English errors that need to be rectified.

Specifics

  1. The authors have failed to place the present work in the context of similar work with shared authors and originating from the same institutions:

(i) Basing LAW, Simpson SV, Adu-Sarkodie Y, Linnes JC. A Loop-Mediated Isothermal Amplification Assay for the Detection of Treponema pallidum subsp. pertenue. Am J Trop Med Hyg. 2020;103(1):253-259. doi:10.4269/ajtmh.19-0243

(ii) Munson M, Creswell B, Kondobala K, et al. Optimising the use of molecular tools for the diagnosis of yaws. Trans R Soc Trop Med Hyg. 2019;113(12):776-780. doi:10.1093/trstmh/trz083.

I do not understand the reason for this omission but it must be rectified.

  1. The asterisks in Table 4 should be explained in a legend.
  2. There is a discrepancy between the description of primers as ‘Only primers FP3 + RP2 for TP and FP3 + RP1 for HD (Table 1) were able to amplify down to 10 DNA molecules/reaction (Fig. 1a), and the primers given in Table 1.
  3. Lines 150, 151 – the BLAST homologies should be clarified instead of referring as ‘high’.
  4. Lines 111-112 – the approval reference number and date should be specified
  5. Line 84 – English to correct

Round 2

Reviewer 2 Report

A brief summary

The authors presented a valuable study with potential to deal with an important public health issue and to aid the WHO-initiative in total eradication of jaws.

General comments 

The paper is now much easier to read and understand.

I find that “eradication efforts” should be (throughout the manuscript) replaced with “eradication program” – it reads better.

Please use the abbreviation “TPHD-RPA” uniformly throughout the manuscript. Please, omit “duplex” – i.e. “duplex TPHD-RPA”.

The authors should address some further aspects. Concerns and suggestions regarding different segments of the manuscript are below.

Abstract

Please mention before line 24 that disease is characterized with chronic skin ulcers.

Line 29 – “Compared…” – please rewrite.

Introduction

The issue of DNA extraction has to be addressed in the Introduction.

Line 40 – “Yaws…” verb is missing (i.e. Yaws is). Please, set commas.

Line 48 – “initial mass…” every person in the endemic community received azithromycin?

Line 58 – “In..” – please rewrite.

Line 66 – “antigens” – antigens, is accurate…

Line 68 – “low” – something is missing.

Line 69 – “Additionally…” – please, rewrite.

Line 73 – “…mass drug administration” = mass drug therapy?

Line 74 – i.e. technique, …(PCR), … address the issue of yaws diagnosis, since it provides high sensitivity and…

Line 77 – MDA – unexplained Abbreviation – write it out.

Line 80 – “The…” – please, set commas.

Line 85 – “6” – six.

Line 87 – “Recombinase” – i.e. recombinase… “tool” – tool, … “and” – and it is less…

Line 93 – “…the field or point-of-need” – is something missing?

Line 94 – “This…” – please rewrite (i.e. stages… stages).

Materials and Methods

Line 102 – “Samples…” – the abbreviations of the districts in Ghana are not necessary.

Line 108 – “Swab…” – please, set commas.

Line 112/113 – The abbreviations KCCR and NMIMR are not necessary.

Line 126 - The abbreviation PPS is not necessary.

Line 157 – “3” – three.

Line 158 – “”design” – designed.

Line 159 – “Mega7” – manufacturer, country?

Line 163 – “3” – three. “3” – three.

Line 173 – “amplitude” – unclear.

Line 176 – “reaction” – unclear, i.e. reaction volume was 50 µl and contained…

Line 183 – “assay or cross…” – unclear.

Line 187 – “49…” – please, rewrite.

Line 193 – “General… - please, rewrite.

Line 196 – “assay” – i.e., assay, …

Results

Line 200 – “assay” – pathogen?

Line 201 – “Only…” – please, rewrite. It is not necessary to mention which primers did well (i.e. FP3, RP2). Without reading the supplemental material, it is not clear what these (unexplained) abbreviations mean. Please, correct the Table 1 accordingly.

Line 202 – “Other…” – omit this sentence.

Line 205 – “Using…” – please, rewrite. The sentence is partly repetitive (the sentence in line 201). “hemolytic cytotoxin” – is this information crucial? It was not mentioned in Materials and Methods.

Line 209 – “One…” – it is unclear, what the authors meant. The results for one DNA copy were not similar for HD and TP? Says the figure legend something else?

Line 220 – “file 1” – file 2.

Line 227 – please, write results clearly - i.e. ulcers (N=35, 71 %), etc.

Line 229 – “Of … - please, rewrite, i.e. “Using qPCR…

Line 231/232 – “one single” – rewrite. “5” – five. The percentage (10%) relates to?

Line 232 – “Two…” – please, consider writing it as follows: “The samples positive for both pathogens by qPCR…”

Line 234 – “Table 3” – Table 2.

Line 238 – “All 18…” – this negative samples were not mentioned in the Table 2.

Table 2. – The Table has to be re-done, in order to be easily understandable. The abbreviations have to be explained. I am not sure if all numbers for specificity, PPV and NPV are necessary for the reader. The “total samples” are mentioned three times – it is confusing. The first part: 16 + 20 is not 31 – confusing. Consider omitting this part. Without information from the text is not clear, how “both pathogens” – twice “6” equals 7. Maybe explain in the legend that 5 + 1 =6, 5 + 1’ = 6.

Discussion

Line 243 – “…ducreyi” – i.e., ducreyi, …

Line 245 – “Molecular…” – please, rewrite and set commas. “…enabling the ability of determination…” – please, rewrite.

Line 251 – LOD – unexplained abbreviation. Not necessary?

Line 252/253 – please, set commas.

Line 254 – “…for” – consider writing, as follows: “of TPDH-RPA assay could provide… ducreyi in lesions, especially in areas where,,, reported.”

Lines 258-264 – “However…” – please, rewrite and shorten sentences, there are repetitive.

Line 267 – “Duplex TPDH-LAMP” assay was not mentioned in the Introduction.

Line 268 – “A lateral…” – what is that? Please, explain (in the Introduction?). Rewrite the sentence.

Line 270 – “The…” – the issue of sensitivity and DNA extraction has to be mentioned in the Introduction.

Line 272 - “These…” – this relates to the information in line 84? Were those results from TPDH-LAMP?

Line 275 – “compared to…” – something is missing – time (minutes)?

Line 276 – “Also…” – please, rewrite. dNTP – unnecessary abbreviation.

Line 280 – “In addition,” – address this issue in the Introduction. State clearly, that in this study was used DNA extracted with “standard/expensive” extraction kits in a well-equipped laboratory. It should be mentioned that a novel TPDH-RPA assay has to be tested with mentioned “less labor-intensive” techniques. Therefore, the conclusion/suggestion in line 286 is too far-fetched.

Line 287 – “Y” – y.

Line 290 – “This…” – please, rewrite.

Conclusion

Line 294 – “Y” – y.

Line 294 – “The…” – please, offer balanced conclusion remarks – i.e. the presented results used DNA extracted with sophisticated extraction kits in a reference laboratory.

Round 3

Reviewer 2 Report

Review Report

Multiplex Recombinase Polymerase Amplification Assay for Simultaneous Detection of Treponema pallidum and Haemophilus ducreyi in yaws-like lesions by Frimpong et al.

A brief summary

The authors presented a valuable study with potential to deal with an important public health issue and to aid the WHO-initiative in total eradication of jaws.

General comments 

The article paper is now much easier to read and understand. However, there are still several points which have to be addressed.

Concerns and suggestions regarding different segments of the manuscript are below.

Abstract

Line 31 – please, omit “confirmed samples”.

Line 33 – “effective” (vs. ineffective) – consider writing “reliable”. “diagnosis” – consider: diagnosis, enable monitoring…

Introduction

Line 41 – “Yaws…”  - verb is missing.

Line 59 – “Patients… - consider: “In patients…” Please, rewrite the sentence.

Line 64 – “These tools…” – consider: “These tests are encumbered with…”

Line 64 – Please omit: “in jaws endemic communities”. I.e. it is applicable “everywhere”.

Line 73 – “These…” – consider writing: “This highlights the need for a sensitive…”

Line 75 – “Nucleic… - consider writing: “…techniques, especially…(PCR), can address the issue of diagnosis of jaws, due to their high sensitivity…”

Line 77 – “This…” – consider writing: “DNA-based diagnostic tests…”. Please omit: “in jaws endemic communities”. I.e. it is applicable “everywhere”.

Line 79 – “…this…” – I guess, that it relates to the PCR. Please, state clearly.

Line 84 – please, omit “detection have”.

Line 85 – “crude extracted…” – I cannot access the referenced article. In the Abstract was written: “The sensitivity of the LAMP assay using unextracted and DNA extracted samples…”. Please, define what “crude extracted” means.

Line 86 – “LAMP… It…” – please, define does this relates to the LAMP used in [15] or in general and rewrite the sentences accordingly. Also, it reads that these pathogens are highly variable under field conditions. Vs. lab conditions, or? Please, rewrite.

Line 73 – “In…” – consider writing: “In contrast… tool. Compared to LAMP, RPA yields… time (30-60 min vs, 15 min, respectively)” … and so on.

Materials and Methods

Line 109 – I would suggest to write what kind of assay that was, i.e. “immunochromatographic assay for the qualitative detection of antibodies of all isotypes (IgG, IgM, IgA) against Treponema pallidum (TP).” (from online description of the assay). The name of the kit could be written in the brackets, without explanation of the abbreviations.

Line 114 – “into” – in. Please, write: “Medicine (Kumasi, Ghana) or […] Research (Accra, Ghana).”

Line 117 – “This…” – please, rewrite the sentence and, i.e. write two sentences instead. Add: “(Accra, Ghana).”

Line 122 – “The…” - verb is missing.

Line 142 – “AmpliMix” – please, describe or spell it out.

Line 149 – In the study were used also separate standards, i.e. just for TP and just for HD. Please, mention that here.

Line 153 – it should write: “dilutions”

Line 173 – “To be…” – please, rewrite the sentence. Please, write it clearly, i.e. include the information from the cover letter (“This is a characteristic of an amplification curve after gradient (exponential) amplification.”)

Line 177 – “kits…” – consider writing: “kit, with minor modifications of the (or according to the) manufacturer’s” …

Line 177 – “template…” – consider writing: “extract”

Line 182 – Please, write “TPHD-RPA… using serial dilutions”. Consider writing: “in eight separate reactions in triplicate.”

Line 184 – “verse” is a wrong word. “log” is also abbreviation. Please, rewrite.

Line 185 – Please, write TPHD-RPA. Omit “as template”.

Line 190 – Omit the abbreviation KCCR. Please, consider writing: “…results of the TPHD-RPA were compared with results of T…”

Line 193 – I guess that the descriptive statistics (Line 196-197) was obtained from Excel. Please, state, if it was so.

Line 195 – Please, consider writing: “qPCR and TPDH-RPA.”

Results

Line 201 – The subtitle is “Analytical Sensitivity and Specificity of TPHD-RPA Assay”, thus, here (Line 202-222) were only presented the data from single-plex RPA assays.

Line 201 – Please, mention these standards in Materials and methods.

Line 202/203 - rewrite. As mentioned earlier, it is not necessary to mention which primers did well (i.e. FP3, RP2). Without reading the supplemental material, it is not clear what these (unexplained) abbreviations mean. Please, correct the Table 1 accordingly.

Line 205 – “10 DNA molecules/reaction”. Please rewrite. It reads that assay amplifies “concentration”. “Figure 1a” – Omitting “a”?

Line 207 – “The start points” – it is not an accurate term, since the fluorescence signal was also present earlier. Please, describe either here or in the figure legend, why was the fluorescence intensity “0” at 4th minute (mixing step in 4th minute – please explain).

Line 213 – please, consider writing “T. pallidum (A) and H. ducreyi (B)” and omitting “(a) […] ducreyi”

Line 217 – please write: “… ducreyi, respectively (Figure 2).”

Figure 2 – Please, rename the x-axe.

Line 222 – Please, explain “CI”.

Line 229 – Please, write TPHD-RPA.

Line 236 – In previous version of the article were 18 negative samples mentioned. That number is from the Table 2 not comprehensible. It reads that 32 samples were negative (last column). That leaves 17 samples not mentioned. Please, explain.

Table 2 – Please, write i.e. “as compared to TPHD multiplex qPCR”. The table is confusing. In the first row, 2nd and 8th column have confusingly similar titles. Twice was written “All samples” – 2nd and 5th row. The line “Samples containing both pathogens” is repetitive, although one number is different (NPV column – 84 vs. 85). In the Line 229 were mentioned “five samples” – it is very hard to read from the table, that this five double-positive together with one single-positive, makes six (as in the table) and one negative. Please, try to explain it, i.e. in the figure legend.

Discussion

Line 246 – please, consider writing: “enabling the determination whether yaws like lesion is caused by T. pallidum od H. ducreyi, or both…”

Line 252 – “These…” - please, consider writing: “Therefore, the analytic sensitivity of the TPHD-RPA assay is comparable the sensitivity of (TPDH) multiplex qPCR (100 […]”

Line 254 – Please, write TPHD-RPA. Please, consider writing: “suggest that this assay… lesion, as well as a distinction between these two pathogens.” Consider omitting: “especially…”

Line 268 – “These TPHD-LAMP assays” – was the same assay used? If yes, then it should write: “This TPHD-LAMP assay”.

Line 271 – “unextracted” – vs. Line 86?

Line 272 – “recorded” – Please, write: showed.

Line 272 – “These… these…” – please, rewrite the whole sentence.

Line 275 – “However…” – please, rewrite. Consider the comment to line 73.

Line 284 - “These…”- please, consider writing: “These DNA extraction...”.

Line 287 – Please omit: “adaptable…” – it is repetitive.

Line 292 – “assay” - please, consider writing: “assay in low resource settings.”

Conclusion

Line 297 – “if” - please, consider writing: “when… friendly DNA extraction technique,”

Line 298-300 – consider the suggestions to line 33 (i.e. one do not monitor program failure).

Other sections

Line 410 – it should be written “Böhlken-Fascher”.

Author Response

Point 1: Line 31 – please, omit “confirmed samples”.

Response 1: “Confirmed” samples has been deleted (Line 31).

Point 2: Line 33 – “effective” (vs. ineffective) – consider writing “reliable”. “diagnosis” – consider: diagnosis, enable monitoring…

Response 2: This sentence has been revised (Line 33).

Point 3: Line 41 – “Yaws…”  - verb is missing.

Response 2: This sentence has been revised (Line 41).

Point 4: Line 59 – “Patients… - consider: “In patients…” Please, rewrite the sentence.

Response 4: This sentence has been revised (Line 67)

Point 5: Line 64 – “These tools…” – consider: “These tests are encumbered with…”

Response 5: This sentence has been revised (Line 72).

Point 6: Line 64 – Please omit: “in jaws endemic communities”. I.e. it is applicable “everywhere”.

Response 6: This sentence has been revised (Line 76).

Point 7: Line 73 – “These…” – consider writing: “This highlights the need for a sensitive…”

Response 7: This sentence has been revised (Line 81).

Point 8: Line 75 – “Nucleic… - consider writing: “…techniques, especially…(PCR), can address the issue of diagnosis of jaws, due to their high sensitivity…”

Response 8: This sentence has been revised (Line 83-86).

Point 9: Line 77 – “This…” – consider writing: “DNA-based diagnostic tests…”. Please omit: “in jaws endemic communities”. I.e. it is applicable “everywhere”.

Response 9: This sentence has been revised (Line 85).

Point 10: Line 79 – “…this…” – I guess, that it relates to the PCR. Please, state clearly.

Response 10: This sentence has been revised (Line 86).

Point 11: Line 84 – please, omit “detection have”.

Response 11: “Detection have” has been omitted (92).

Point 12: Line 85 – “crude extracted…” – I cannot access the referenced article. In the Abstract was written: “The sensitivity of the LAMP assay using unextracted and DNA extracted samples…”. Please, define what “crude extracted” means.

Response 12: This sentence has been revised (Line 93).

Point 13: Line 86 – “LAMP… It…” – please, define does this relates to the LAMP used in [15] or in general and rewrite the sentences accordingly. Also, it reads that these pathogens are highly variable under field conditions. Vs. lab conditions, or? Please, rewrite.

Response 13: This sentence has been revised (Line 94-96).

Point 14: Line 73 – “In…” – consider writing: “In contrast… tool. Compared to LAMP, RPA yields… time (30-60 min vs, 15 min, respectively)” … and so on.

Response 14: Thanks for the suggestion but authors would like this sentence to remain as it is (Line 88).

Point 15: Line 109 – I would suggest to write what kind of assay that was, i.e. “immunochromatographic assay for the qualitative detection of antibodies of all isotypes (IgG, IgM, IgA) against Treponema pallidum (TP).” (from online description of the assay). The name of the kit could be written in the brackets, without explanation of the abbreviations.

Response 15: This sentence has been revised (Line 108-112).

Point 16: Line 114 – “into” – in. Please, write: “Medicine (Kumasi, Ghana) or […] Research (Accra, Ghana).”

Response 16: This sentence has been revised (Line 116-117).

Point 17: Line 117 – “This…” – please, rewrite the sentence and, i.e. write two sentences instead. Add: “(Accra, Ghana).”

Response 17: This sentence has been revised (Line 120-122).

Point 18: Line 122 – “The…” - verb is missing.

Response 18: This sentence has been revised (Line 124).

Point 19: Line 142 – “AmpliMix” – please, describe or spell it out.

Response 19: AmpliMix is a proprietary name by the manufacture to its master mix reagent (Line 144).

Point 20: Line 149 – In the study were used also separate standards, i.e. just for TP and just for HD. Please, mention that here.

Response 20: The molecular DNA standard was a single plasmid made up of a fragment of 300 bp T. pallidum PolA gene and another fragment 300 bp of H. ducreyi hemolytic cytotoxin HhdA gene (Line 151).

Point 21: Line 153 – it should write: “dilutions”

Response 21: This sentence has been revised (Line 155).

Point 22: Line 173 – “To be…” – please, rewrite the sentence. Please, write it clearly, i.e. include the information from the cover letter (“This is a characteristic of an amplification curve after gradient (exponential) amplification.”)

Response 22: This sentence has been revised (Line 176).

Point 23: Line 177 – “kits…” – consider writing: “kit, with minor modifications of the (or according to the) manufacturer’s” …

Response 23: This sentence has been revised (Line 179).

Point 24: Line 177 – “template…” – consider writing: “extract”

Response 24: This sentence has been revised (Line 182).

Point 25: Line 182 – Please, write “TPHD-RPA… using serial dilutions”. Consider writing: “in eight separate reactions in triplicate.”

Response 25: This sentence has been revised (Line 184-185).

Point 26: Line 184 – “verse” is a wrong word. “log” is also abbreviation. Please, rewrite.

Response 26: This sentence has been revised (Line 186).

Point 27: Line 185 – Please, write TPHD-RPA. Omit “as template”.

Response 27: This sentence has been revised (Line 188).

Point 28: Line 190 – Omit the abbreviation KCCR. Please, consider writing: “…results of the TPHD-RPA were compared with results of T…”

Response 28: This sentence has been revised (Line 193).

Point 29: Line 193 – I guess that the descriptive statistics (Line 196-197) was obtained from Excel. Please, state, if it was so.

Response 29: Descriptive statistics was obtained with GraphPad Prism v.8.

Point 30: Line 195 – Please, consider writing: “qPCR and TPDH-RPA.”

Response 30: This sentence has been revised (Line 198).

Point 31: Line 201 – The subtitle is “Analytical Sensitivity and Specificity of TPHD-RPA Assay”, thus, here (Line 202-222) were only presented the data from single-plex RPA assays.

Response 31: This sentence has been revised (Line 204).

Point 32: Line 201 – Please, mention these standards in Materials and methods.

Response 32: The molecular DNA standard has been mentioned in line 151-153.

Point 33: Line 202/203 - rewrite. As mentioned earlier, it is not necessary to mention which primers did well (i.e. FP3, RP2). Without reading the supplemental material, it is not clear what these (unexplained) abbreviations mean. Please, correct the Table 1 accordingly.

Response 33: Table 1 has been revised and a legend explaining these abbreviations have been stated (Line 209-212)

Point 34: Line 205 – “10 DNA molecules/reaction”. Please rewrite. It reads that assay amplifies “concentration”. “Figure 1a” – Omitting “a”?

Response 34: This sentence has been revised (208).

Point 35: Line 207 – “The start points” – it is not an accurate term, since the fluorescence signal was also present earlier. Please, describe either here or in the figure legend, why was the fluorescence intensity “0” at 4th minute (mixing step in 4th minute – please explain).

Response 35: This sentence has been revised (Line 213-214) and the reason for fluorescence intensity of “0” at 4th minutes stated in the legend (220-221).

Point 36: Line 213 – please, consider writing “T. pallidum (A) and H. ducreyi (B)” and omitting “(a) […] ducreyi”

Response 36: This sentence has been revised (Line 221-222)

Point 37: Line 217 – please write: “… ducreyi, respectively (Figure 2).”

Response 37: This sentence has been revised (Line 224).

Point 38: Figure 2 – Please, rename the x-axe.

Response 38: The name of the x-axes should remain unchanged.

Point 39: Line 222 – Please, explain “CI”.

Response 39: “CI” Has been explained in the legend (Line 229).

Point 40: Line 229 – Please, write TPHD-RPA.

Response 40: This sentence has been revised (Line 232).

Point 41: Line 236 – In previous version of the article were 18 negative samples mentioned. That number is from the Table 2 not comprehensible. It reads that 32 samples were negative (last column). That leaves 17 samples not mentioned. Please, explain.

Response 41: In determining, the specificity of the TPHD assay for T. pallidum as an example, samples containing only H. ducreyi (14) were added to samples negative for both pathogens (18), since these samples will be negative for T. pallidum demonstrating 100% specificity of the TPHD-RPA assay to T. pallidum detection. This makes samples negative for T. pallidum being thirty-two (32) instead of eighteen (18).   

Point 42: Table 2 – Please, write i.e. “as compared to TPHD multiplex qPCR”. The table is confusing. In the first row, 2nd and 8th column have confusingly similar titles. Twice was written “All samples” – 2nd and 5th row. The line “Samples containing both pathogens” is repetitive, although one number is different (NPV column – 84 vs. 85). In the Line 229 were mentioned “five samples” – it is very hard to read from the table, that this five double-positive together with one single-positive, makes six (as in the table) and one negative. Please, try to explain it, i.e. in the figure legend.

Response 42: Column names for table 2 have been revised. The diagnostic performance for the TPHD-RPA assay was determine for each other organism.  The NPV is different because the number of true negatives is different for both organisms. For the seven (7) samples positive for both pathogens using qPCR, 6 were positive for TP, 6 HD and 5 for both TP and HD using the TPHD assay. 

Point 43: Line 246 – please, consider writing: “enabling the determination whether yaws like lesion is caused by T. pallidum od H. ducreyi, or both…”

Response 43: This sentence has been revised (Line 252-254).

Point 44: Line 252 – “These…” - please, consider writing: “Therefore, the analytic sensitivity of the TPHD-RPA assay is comparable the sensitivity of (TPDH) multiplex qPCR (100 […]”

Response 44: Thanks for the suggestion but based on this revised manuscript, this sentence should remain as it is (Line 257-258).

Point 45: Line 254 – Please, write TPHD-RPA. Please, consider writing: “suggest that this assay… lesion, as well as a distinction between these two pathogens.” Consider omitting: “especially…”

Response 45: This sentence has been revised (Line 261-263).

Point 46: Line 268 – “These TPHD-LAMP assays” – was the same assay used? If yes, then it should write: “This TPHD-LAMP assay”.

Response 46: The TPHD LAMP assay were different (Line 272-274).

Point 47: Line 271 – “unextracted” – vs. Line 86?

Response 47: Line 86 (now Line 85) has been revised.

Point 48: Line 272 – “recorded” – Please, write: showed

Response 48: This sentence has been revised (Line 278).

Point 49: Line 272 – “These… these…” – please, rewrite the whole sentence.

Response 49: The diagnosis performance of TPHD- RPA and LAMB have been already stated (Line 275 and 278).

Point 50: Line 275 – “However…” – please, rewrite. Consider the comment to line 73.

Response 50: Thanks for the suggestion but authors would like this sentence to remain as it is (Line 281-283).

Point 51: Line 284 - “These…”- please, consider writing: “These DNA extraction...”.

Response 51: This sentence has been revised (Line 290).

Point 52: Line 287 – Please omit: “adaptable…” – it is repetitive.

Response 52: This sentence has been revised (Line 293).

Point 53: Line 292 – “assay” - please, consider writing: “assay in low resource settings.”

Response 53: This sentence has been revised (Line 293).

Point 54: Line 297 – “if” - please, consider writing: “when… friendly DNA extraction technique,”

Response 54: This sentence has been revised (Line 302).

Point 55: Line 298-300 – consider the suggestions to line 33 (i.e. one do not monitor program failure).

Response 55: This sentence has been revised (Line 303-304).

Point 56: Line 410 – it should be written “Böhlken-Fascher”.

Response 56: This sentence has been revised (Line 412).
